# Recent Progress on the Localization of the Spindle Assembly Checkpoint Machinery to Kinetochores

**DOI:** 10.3390/cells8030278

**Published:** 2019-03-23

**Authors:** Zhen Dou, Diogjena Katerina Prifti, Ping Gui, Xing Liu, Sabine Elowe, Xuebiao Yao

**Affiliations:** 1Anhui Key Laboratory of Cellular Dynamics & Chemical Biology, Hefei National Science Center for Physical Sciences at the Microscale, School of Life Sciences, University of Science & Technology of China, Hefei 230027, China; guiping@mail.ustc.edu.cn (P.G.); xing1017@ustc.edu.cn (X.L.); 2Programme in Molecular and Cellular Biology, Faculty of Medicine, Université Laval, 1050 Avenue de la Medecine, Bureau 4633, Universite Laval, Quebec, QC G1V0A6, Canada; kprifth@gmail.com (D.K.P.); Sabine.Elowe@crchudequebec.ulaval.ca (S.E.); 3Keck Center for Cellular Dynamics & Organoids Plasticity, Morehouse School of Medicine, Atlanta, GA 30310, USA; 4Centre de Recherche de CHU de Québec, Université Laval, 1050 Avenue de la Medecine, Bureau 4633, Universite Laval, Quebec, QC G1V0A6, Canada; 5Department of Pediatrics, Faculty of Medicine, Université Laval, 1050 Avenue de la Medecine, Bureau 4633, Universite Laval, Quebec, QC G1V0A6, Canada; 6CAS Center of Excellence on Molecular Cell Sciences, Hefei 230027, China

**Keywords:** spindle assembly checkpoint (SAC), kinetochore, microtubule, kinase, localization

## Abstract

Faithful chromosome segregation during mitosis is crucial for maintaining genome stability. The spindle assembly checkpoint (SAC) is a surveillance mechanism that ensures accurate mitotic progression. Defective SAC signaling leads to premature sister chromatid separation and aneuploid daughter cells. Mechanistically, the SAC couples the kinetochore microtubule attachment status to the cell cycle progression machinery. In the presence of abnormal kinetochore microtubule attachments, the SAC prevents the metaphase-to-anaphase transition through a complex kinase-phosphatase signaling cascade which results in the correct balance of SAC components recruited to the kinetochore. The correct kinetochore localization of SAC proteins is a prerequisite for robust SAC signaling and, hence, accurate chromosome segregation. Here, we review recent progresses on the kinetochore recruitment of core SAC factors.

## 1. Introduction

During cell division, it is crucial to transmit the duplicated genome to two daughter cells equally. Kinetochores, the large protein complexes that assemble at the centromere of a chromosome, are an essential protein machinery that orchestrates faithful mitosis. Kinetochores function to align sister chromatids in prometaphase and to pull the sister chromatids apart in anaphase by connecting the chromosomes to microtubules of the spindle. Another important function of the kinetochore is to initiate Spindle Assembly Checkpoint (SAC) signaling. The SAC restrains cells from entering anaphase until all sister chromatids are attached to the microtubules radiating from the two opposite poles of the mitotic spindle. Biochemically, the SAC generates a diffusible, cytoplasmic mitotic checkpoint complex (MCC), a so-called “wait anaphase” signal, that ultimately results in the inhibition of the E3 ubiquitin ligase Anaphase Promoting Complex/Cyclosome (APC/C), hence prolonging the mitotic state [1,2]. During prophase, the checkpoint kinase Monopolar spindle 1 (Mps1) phosphorylates its substrates to activate the SAC [3], while Aurora B kinase, the catalytic subunit of chromosome passenger complex (CPC), phosphorylates the targets at improperly attached kinetochores to promote error correction [4]. The phosphorylation of the N-terminal tail region of Nuclear division cycle 80/High expression in cancer 1 (Ndc80/Hec1) by Aurora B leads to the recruitment of Mps1 at the kinetochore [5,6,7]. Mps1 phosphorylates the outer kinetochore protein Kinetochore null 1 (Knl1) at several Met-Glu-Leu-Thr (MELT) motifs, creating docking sites for the recruitment and formation of the MCC which consists of mitotic arrest deficient 2 (Mad2), budding uninhibited by benzimidazoles related 1 (BubR1), budding uninhibited by benzimidazole 3 (Bub3) and cell division cycle 20 (Cdc20) [3,8,9,10]. The Bub3-Bub1 complex recognizes and docks at these phosphorylated MELT sequences. This complex will then recruit the Bub3-BubR1 complex, Cdc20 and Mad-Mad2. Mad1-Mad2 catalyzes the conversion of Mad2 from an inactive, open-like conformation to an active, closed-like conformation, thereby favoring the formation of the MCC [3]. In prometaphase, after nuclear envelope breakdown, at least two different phosphatases, PP1 and PP2A, are recruited at the kinetochores via various adaptor subunits to counteract kinase activity and to locally silence the SAC [11,12].

As indicated above, the major components of this surveillance mechanism, originally identified in budding yeast, include Mad1, Mad2, Mad3 (the budding yeast orthologue of BubR1), Bub1, Bub3 [13,14] and Mps1 [15,16]. In addition, numerous data suggest the Aurora B is also a core component of the SAC [3,5,17]. These proteins are well-conserved throughout evolution [18,19]. One striking exception is the complete absence of an Mps1 orthologue in C. elegans, and it has been hypothesized that, in this organism, the functions of Mps1 may be fulfilled by other kinases such as Polo-like kinase 1 (Plk1) [20]. In addition, the evolutionary paths of Mad3 and its orthologue BubR1 appear to have diverged considerably; BubR1 in higher eukaryotes has a kinase domain that, at least in vertebrates, appears to have degenerated into a pseudokinase [21]. The SAC is more intricately regulated in higher eukaryotes, and additional components to the core SAC module have been shown to contribute to the SAC in mammalian cells. In addition to these shared core SAC components, Rod, ZW10 and Zwilch (collectively known as the RZZ complex) are also essential for the robust SAC function in metazoans [2], as is the microtubule attachment machinery including the KMN network (consisting of the Knl1-Mis12-Ndc80 subcomplexes) that promotes microtubule binding through the Ndc80/Hec1 subunit of the Ndc80 complex [22,23,24,25]. Moreover, in vertebrate cells, Cdc20 is thought to play dual functions. On the one hand, as a component of MCC, it works as an anaphase inhibitor; on the other hand, it promotes the metaphase-anaphase transition as an activation factor of the APC/C [3,26,27]. Once attachment is achieved, the SAC must be extinguished in order to allow mitotic exit. The factors involved in silencing the SAC signaling include p31^Comet^, Trip13 and the phosphatases PP1 and PP2A. Detailed discussions of SAC silencing and the critical but complicated function of CDK1-Cyclin B in mitosis are beyond the scope of this review, and the reader is referred to excellent recent reviews on these topics [3,11,12,28].

The kinetochore functions not only as a structural platform bridging mitotic chromosomes and microtubules but also as a major mitotic signaling hub and is tasked with coordinating the microtubule attachment with the SAC activity [29]. With the exception of the inner centromere localization of Aurora B and its regulatory partners Inner Centromere Protein (INCENP), Survivin and Borealin (which collectively form the CPC), the other core SAC components localize at the outer kinetochore and fibrous corona. Kinetochore localization of the spindle checkpoint kinases appears to be required for their optimal activation and function. Mps1 kinetochore docking has been proposed to partially compete for microtubule binding at the kinetochores in mammalian cells [30,31,32]. In budding yeast, kinetochore attachment to the microtubule has been suggested to physically separate Mps1 from its downstream substrates, effectively terminating SAC signaling [33]. Another example is chromosome-bound Bub1 that it is hyperphosphorylated and activated at unattached kinetochores [34]. Bub1 autophosphorylation has also been shown to regulate its turnover at kinetochores and, thus, its access to its major downstream substrate, Histone H2A [35]. The intricate workings of kinetochore targeting of these SAC proteins are gradually becoming elucidated. In this article, we will review the molecular pathways of kinetochore recruitment of these core SAC proteins.

## 2. The KMN Network Is the Structural Scaffold of the Outer Kinetochore

It is generally accepted that the kinetochore localization of SAC proteins is critical for proper checkpoint signaling despite some evidence showing MCC presence and activation in interphase cells [36,37,38,39,40,41,42]. Indeed, Mps1 activity is required for the assembly of the Cdc20 inhibitory complex in interphase and mitosis [43], and it has been proposed that the nuclear pore complex can act as a scaffold for generating the anaphase inhibitory signal in interphase [44]. In addition, Malureanu et al. also showed that kinetochore-bound BubR1 is nonessential and that soluble BubR1 functions as a pseudosubstrate inhibitor of APC/C during interphase [45]. Therefore, exploring the kinetochore targeting of SAC proteins is of great importance to understand the molecular details of the initiation, propagation and termination of SAC signaling. In this context, the KMN network has emerged as an essential docking platform for SAC proteins [40].

In human cells, sixteen kinetochore proteins termed the Constitutive Centromere-Associated Network (CCAN) reside at centromeric DNA through binding with the CENP-A nucleosome throughout the cell cycle [46]. These then serve as the targeting factors for the outer kinetochore which is assembled only during mitosis (Figure 1). The core outer kinetochore proteins comprise a conserved network called KMN, including the Knl1-Zwint1 subcomplex, the Mis12 complex (Mis12C, which contains Mis12, Dsn1/Mis13, Nsl1/Mis14 and Nnf1) and the Ndc80 complex (Ndc80C, which contains Hec1/Ndc80, Nuf2, Spc24 and Spc25) [47,48]. It is now well-established that the outer kinetochore KMN network is the major interface connecting spindle microtubules to kinetochores [23,24,25,47,48,49]. Among the KMN network components, the N-terminal regions of Hec1 and Nuf2 that both contain a globular CH domain are thought to be the major microtubule binding sites [22,23,25]. Knl1 and the Mis12C also contribute to the synergistic microtubule-binding activity of the KMN network [23,50,51]. 

Besides a globular CH domain, Hec1 harbours an unstructured, positively charged N-terminal tail. A significant body of evidence has shown that the N-terminal tail has a high binding affinity to microtubules through electrostatic interaction and is critical for kinetochore-microtubule attachment and chromosomal alignment [24,25,52,53]. Importantly, the phosphorylation of Hec1 N-terminal tail by Aurora B destabilizes the kinetochore-microtubule attachment and promotes error correction in early mitosis [22,23,51]. A pioneering work showed that yeast Ndc80 is a substrate of Ipl1/Aurora B and proposed that the phosphorylation of Dam1 and Ndc80 by Ipl1 leads to destabilized kinetochore-microtubule attachment [54]. In addition, phosphorylated Ndc80 clearly reduced the binding affinity with microtubules in vitro [23]. In parallel, it was confirmed that Aurora B phosphorylates human Hec1 N-tail and the the non-phophorylatable Hec1-6A mutant displayed severe chromosome alignment defects [22]. Interestingly, the Nogales group demonstrated that Hec1 N-tail mediates the oligomerization of Ndc80C [49]. They proposed that the phosphorylation of Hec1 N-tail by Aurora B prevents the oligomerization and clustering of Ndc80C along microtubule. Recently, our group demonstrated that acetyltransferase Tip60 acetylates Hec1 at two evolutionarily conserved residues, Lys53 and Lys59, and that this acetylation weakened the phosphorylation of Hec1 N-tail at Ser55 and Ser62 [55]. In conclusion, the Ndc80/Hec1 N-tail plays a crucial role in regulating kinetochore-microtubule attachment. After phosphorylation by Aurora B, the reduced binding between Ndc80C and microtubules contributes to the removal of incorrect microtubule attachments.

The KMN network and other outer kinetochore proteins are rapidly assembled on the inner kinetochore when cells enter mitosis. Recent work has highlighted two parallel pathways for the assembly of the outer kinetochore and the KMN network. The N-terminal region of Centromere protein C (CENP-C) recruits Mis12C through direct binding between the N-terminal region of CENP-C and Mis12C [56,57]. In turn, Mis12C can recruit Knl1 (in complex with Zwint1) and Ndc80C [46,58]. The other pathway depends on the CENP-TWSX subcomplex, which can bind centromeric DNA. Centromere protein T (CENP-T) can subsequently recruit Ndc80C via the direct binding between the CENP-T N-terminal tail and the Spc24/Spc25 subunit of Ndc80C upon mitotic entry [59,60]. Interestingly, targeting CENP-C-ΔC-LacI and CENP-T-ΔC-LacI to the ectopic chromosomal LacO position endows the CENP-C/CENP-T foci with the ability to both bind microtubules and to recruit multiple outer kinetochore proteins, indicating that CENP-C and CENP-T are sufficient targeting factors for the formation of a functional outer kinetochore [61].

The mature KMN network assembles in prophase, and several mechanisms determine the timing of KMN assembly. The Ndc80C is sequestered outside the nucleus throughout interphase and is thereby spatially separated from the CCAN until mitosis after nuclear envelope breakdown [62]. Second, CDK1 and Aurora B, which are largely inactive outside mitosis, play a critical function in promoting KMN assembly. The CDK1 phosphorylation of CENP-T promotes the direct interaction of CENP-T with Ndc80C and Mis12C (Figure 1A) [62,63,64,65]. In addition, the phosphorylation of Dsn1/Mis13 by Aurora B enhances the interaction between CENP‑C and the Mis12C during mitosis (Figure 1B) [66,67,68], echoing our early findings that Aurora B phosphorylation towards Dsn1/Mis13 is essential for the assembly of functional outer kinetochore [69]. Very recently, a study from the Fukagawa group investigated the contribution of the CENP-C and CENP-T pathways in the recruitment of the KMN network to kinetochores and concluded that the CENP-T pathway plays a dominant role in loading the KMN network in chicken DT40 cells [70]. This study suggests the existence of a remarkable phosphorylation-regulated plasticity in the inner–outer kinetochore interface during different mitotic stages and in different species/cells.

## 3. The Kinetochore Recruitment of Mps1 Depends on Ndc80C and Aurora B Activity

Mps1 lies at or near the apex of SAC signaling and, as such, is one of the first SAC components to be recruited to kinetochores. The recruitment of downstream SAC components requires Mps1 kinase activity, and therefore, Mps1 is considered an initiator of checkpoint signaling [71]. Disrupting the Mps1 function through either knocking down the Mps1 protein or chemically inhibiting its kinase activity results in premature mitotic exit with aberrant segregation of the sister chromatids [72,73,74]. The sufficient, localized kinase activity of Mps1 is, thus, stringently required for the functional integrity of SAC. Mps1 also participates in chromosome alignment and error correction, but the exact mechanism remains elusive [71]. A recent work demonstrated that Mps1 corrects the erroneous attachment by phosphorylating the Ska complex and, thereafter, by destabilizing the microtubule attachment [75]. 

Mps1 kinetochore localization remains poorly understood. Human Mps1 harbours an N-terminal extension (NTE) followed by a tetratricopeptide repeat (TPR) domain [6,76]. The NTE and MR (Middle Region, a short stretch in the middle of the protein) collectively coordinate Mps1 kinetochore docking [6,32]. In addition, the N-terminus (aa. 1–300) of Mps1 fine-tunes the catalytic activity [6,76]. The dimerization-induced autophosphorylation of the NTE results in the robust activation of Mps1 through the relief of autoinhibition [77]. Mounting evidences indicate that Hec1 is required for the kinetochore localization and effective activation of Mps1 [5,6,7,78,79]. The Kops and Yu groups demonstrated a contribution of the Hec1 N-terminal region for Mps1 kinetochore localization and provided the evidence for a direct interaction between human Hec1 and Mps1 [30,31,32]. The direct binding of Mps1 at the CH domains of Hec1 through the NTE and of Nuf2 through the MR suggest a competition with microtubules. In this model (Figure 2A), the formation of stable end-on attachments would essentially obscure the Mps1 binding site, leading to Mps1 displacement from the kinetochore and SAC silencing [30,31]. Upon microtubule attachment with Ndc80C, Mps1 binding to kinetochores is excluded and SAC signaling is quenched. The competitive binding of Ndc80C with microtubules and Mps1 provides a mechanistic explanation for how end-on microtubule binding to kinetochores couples SAC signaling to its silencing [80,81]. Our group also demonstrated that the aberrant high-affinity kinetochore localization of inactive Mps1 interferes with the establishment of stable kinetochore-microtubule attachment, supporting the competition between Mps1 and microtubules with Ndc80C [32]. Together, these recent publications support the notion that Ndc80C serves the kinetochore sensor to transmit the microtubule attachment state to the SAC signaling machinery.

As indicated above however, in budding yeast, microtubule attachment to Ndc80 can generate a physical distance between Mps1 and its relative targets which may also contribute to SAC silencing, highlighting the importance of sensing both attachment and tension [33,80,81]. Indeed while attractive, the competitive binding model needs to be reconciled with the evidence that Mps1 can localize at kinetochores with microtubule attachment, when preventing the relocation of CPC to the central spindle in anaphase [82]. Consistent with the comparable Mps1 levels at kinetochores assembled in *Xenopus* egg extracts treated with nocodazole or STLC, which induces syntelic microtubule attachment, we also observed clear Mps1 localization at kinetochores in HeLa cells arrested by Monastrol treatment [32,83]. It is very likely that a syntelic attachment might not all be end-on. There is also evidence that error correction, such as lateral to end-on conversion, requires multiple factors (i.e., Centromere protein E/CENP-E and Mitotic centromere-associated kinesin/MCAK) [84]. In addition, Isokane et al. implied a role for ARHGEF17, a Rho family GTPase exchange factor protein, in the SAC through targeting Mps1 to mitotic kinetochores independently of its Rho GEF catalytic activity [85]. Based on this study, phosphorylated ARHGEF17 forms a complex with inactive Mps1 and localizes it at kinetochores, where Mps1 phosphorylates ARHGEF17 to drive its own release from the kinetochores. Further studies are required for understanding the exact mechanism by which ARHGEF17 drives Mps1 localization and how the interaction between these proteins is established and regulated.

Although the details remain unclear, Aurora B activity promotes efficient Mps1 recruitment to unattached kinetochores, allowing rapid Mps1 activation at the onset of mitosis [5,86,87,88]. During prophase, Mps1 acts as the initiator of SAC signaling, while Aurora B prevents its substrates from attaching to microtubules. Considering that Aurora B promotes timely Mps1 recruitment and that the Aurora B target Hec1 is a direct receptor for Mps1 kinetochore recruitment, a straightforward hypothesis is that Aurora B enhances Mps1 localization by directly phosphorylating the Hec1 N-tail (Figure 1C). Indeed, Zhu et al. provided evidence to support this idea [7]. Another possibility is Aurora B releases the inhibitory effect of the Mps1 TPR domain on kinetochore localization [6]. Alternatively, it may well be that the spatial separation of Aurora B kinase from its outer kinetochore substrates (such as Ndc80C) upon the end-on microtubule binding and establishment of tension extinguishes Mps1 kinetochore localization and SAC signaling (Figure 2B) [89]. We also note that there is evidence against the spatial separation of Aurora B being important for SAC, as centromeric Aurora B is not required for recruitment of BubR1 and Mad2 to unattached kinetochores [83]. Potentially, both the competitive binding of microtubules and Mps1 with Ndc80C and the spatial separation of Aurora B kinase from the outer kinetochore contribute to coupling kinetochore-microtubule attachment with SAC signaling. Besides Aurora B, Cdk1 phosphorylates Mps1 at Ser281 (and other sites) and potentiates Mps1 activity [88,90]. However, whether this phosphorylation event enhances Mps1 kinetochore localization is unclear [90,91].

## 4. The Recruitment of Bub1/Bub3 and BubR1/Bub3

Bub1 and BubR1 are two key SAC factors. Bub1 and BubR1 are thought to have evolved from a single ancestral gene through a number of independent gene duplication events. They share a similar domain architecture, including an N-terminal tetratricopeptide repeat domain (TPR) and a Bub3-binding domain (B3BD, also known as the GLEBS motif for GLE2p-binding sequence), both of which contribute to their kinetochore localization [92,93,94,95]. Despite considerable sequence similarity and domain organization, Bub1 and BubR1 have distinct functions during mitosis. In keeping with this, they also have distinct mechanisms of recruitment which have been recently elucidated. Bub1 and BubR1 are absolutely dependent on both Knl1 as well as Bub3 for kinetochore tethering, although these proteins contribute differently in each case [93,96,97]. The Mps1 phosphorylation of Knl1 on conserved MELT motifs at the phosphoacceptor. Threonine in yeast and human cells essentially primes the localization of Bub1 and BubR1 (Figure 1D) [8,9,10,98]. Elegant structural biology studies have demonstrated that Bub3 is the reader for phosphorylated MELT motifs [99]. Bub3-bound Bub1 docks onto Knl1 through the direct binding of a structurally conserved interface on the side of the beta-propeller fold of Bub3 to the phophorylated MELT motifs. The binding of Bub3-Bub1 is significantly stronger than the binding of Bub3-BubR1 as a consequence of a loop region N-terminal to the B3BD of Bub1 which enhances the binding of Bub3 with phosphorylated Knl1. In BubR1, the equivalent loop is essential for the SAC, and the ability of the MCC to inhibit APC/C activity. The bulk of BubR1 at kinetochores is likely recruited through direct hetero-dimerization with Bub1 rather than with Knl1 [100,101]. 

Interestingly, there are multiple MELT motifs in Knl1 proteins (up to 19 in human Knl1 and 5 in yeast Knl1). However, not all MELT motifs have the same activity (binding affinity) towards Bub3-Bub1. Ten MELT motifs have an N-terminal TxxΩ motif (x, any amino acid; Ω, aromatic). These TxxΩ-MELT motifs are essential for the efficient recruitment of Bub1/BubR1 [98]. Knl1 N-terminal fragments encompassing MELT motifs followed by and KI motifs (short 12 residue motifs, termed Lys–Ile) are capable of recruiting low levels of Bub1/BubR1 sufficient for SAC function but insufficient for the correction of chromosome alignment [98,102]. Several MELT motifs contain an adjacent downstream SHT (Ser-His-Thr) motif. The phosphorylated MELT will prime the phosphorylation of SHT, also by Mps1. Together, MELpT and SHpT have an enhanced binding affinity with Bub3-Bub1 [103]. In a parallel fashion, the TPRs of Bub1 and BubR1 bind to neighbouring KI motifs at the N-terminus of Knl1 directly and may enhance the binding of Bub3-Bub1 and Bub3-BubR1 to the first MELT motif which lies adjacent [102]. Consistent with the aforementioned studies, the Nilsson group found that a Knl1 truncation with 4 MELT motifs is sufficient to support proper Knl1 function [104]. The evolution of many MELT motifs with varying Bub3-Bub1 recruitment capacities may represent a mechanism through which eukaryotic cells can fine-tune the amount of Bub3-Bub1 that is recruited to kinetochores and, thus, dictate the strength of the SAC. How this fine-tuning occurs, however, remains to be explored.

## 5. The Kinetochore Localization of Mad1/Mad2 Relies on Bub1 and RZZ

Mad2 is a core component of MCC. It forms a constitutive heterotetramer with Mad1 which the main role of is to recruit Mad2 to kinetochores [105,106,107,108,109]. Once at the kinetochores, Mad2 is converted from an open (O-Mad2) to a closed (C-Mad2) conformation, which is critical for its checkpoint activity [110,111]. Numerous data suggest that Mad1/Mad2 are the most downstream components of the SAC [78,112,113,114,115] and Mps1 kinase activity is absolutely required for the kinetochore targeting of both Mad1 and Mad2 [43,87,116]. Although the localization dependency relationship between Mad1 and Mad2 is well-established, the direct kinetochore receptor of Mad1 remained elusive for many years. Recent studies in several model organisms have confirmed that Bub1 is a direct scaffold for Mad1 at the kinetochores. A pioneering work in budding yeast from the Biggins lab demonstrated that the phosphorylation of the middle region of Bub1 by Mps1 is required for the direct recruitment of Mad1 [117]. Soon after this initial finding, the Bub1-Mad1 interaction was shown to be conserved and required for the kinetochore localization of Mad1 in fission yeast, worm and mammalian cells despite the poor conservation of phosphorylation sites in Bub1 homologs [118,119,120]. In short, Knl1 recruits Bub3/Bub1, and then, Bub1 recruits Mad1 (referred to as the Knl1-Bub3-Bub1/KBB pathway). Subsequently, the Yu group demonstrated that the sequential phosphorylation of human Bub1 by Cdk1 and Mps1 enhances Mad1 binding in a manner that is critical for SAC activation in human cells (Figure 1D). In addition, Mps1 also phosphorylates Mad1 itself, and phosphorylated Mad1 directly interacts with Cdc20 [121]. In elegant in vitro reconstitution experiments, the Musacchio group confirmed Mps1 phosphorylates and activates Mad1, ensuring robust SAC response [122].

In addition to the KBB pathway mentioned above, evidence suggests that the Rod-ZW10-Zwilch (RZZ) pathway contributes to targeting Mad1/Mad2 to kinetochores in human cells [123,124]. Rod and ZW10 are key players for SAC function [125,126], whereas Zwint1 links structural kinetochore components such as Knl1 with checkpoint signaling (RZZ complex) [127,128]. Genetic and cell biology experiments in the fruit fly demonstrated that the RZZ complex contributes to checkpoint activation by promoting Mad2 recruitment and to checkpoint inactivation by recruiting dynein/dynactin that subsequently removes Mad2 from attached kinetochores [129]. In human non-transformed diploid RPE1 cells depleted of Knl1, unattached kinetochores can recruit sufficient amounts of Mad1/Mad2 and can hold cells in mitosis in an RZZ-dependent manner. In this situation, the RZZ complex functions by recruiting Mad1/Mad2 to unattached kinetochores [123]. Further studies reveal that the RZZ complex is required for the maintenance of the SAC but not the initiation of SAC signaling. Thus, in early mitosis, Mad1 kinetochore localization transiently depends on Bub1 phosphorylated by Mps1; the RZZ complex subsequently maintains Mad1 kinetochore accumulation in late mitosis, even in the absence of Bub1 [38,130]. Interestingly and in support of these observations, Zhang et al. propose that Bub1 mediates RZZ recruitment at kinetochores, which is in turn required for Mad1 kinetochore localization [101]. In addition, it was proposed that RZZ interacts with the N-terminus of Knl1 independently of Zwint1 to facilitate Mad1 kinetochore localization [123,124]. Recently, using the combination of CRISPR-Cas9 Knockout and siRNA treatment, two studies demonstrated that RZZ’s primary role is to localize Mad1 at kinetochores. Forced Mad1 kinetochore localization can bypass the requirement of RZZ for the SAC function, whereas Bub1 contributes to the SAC beyond its role in recruiting Mad1/Mad2 [130,131]. In support of the function of the RZZ complex in SAC signaling, a study highlights the interaction between RZZ and Mad1, which were shown to co-immunoprecipitate, suggesting that there is a physical interaction [132].

A recent work has highlighted the expandable nature of the outermost layers of the kinetochore, in particular the fibrous corona, which has been proposed to form a relatively stable structure that is regulated in a manner distinct from the outer kinetochore [133,134]. The RZZ complex plays a central role in creating a mesh-like fibrous corona structure, and both the RZZ complex and spindly are essential for kinetochore expansion at unattached human kinetochores. This expandable module includes RZZ complex, Spindly, Dynein-Dynactin, CENP-E and Mad1/Mad2 and is regulated by mitotic kinases including Cdk1 and Mps1 [130,133].

## 6. Concluding Remarks

In essence, SAC is a cellular signaling pathway. Multiple mitotic kinases and their substrates are involved in this signaling. Therefore, the correct position of specific kinases to its substrates is of great importance for the functional integrity of the SAC. We envision the kinetochore localization of SAC factors may serve several functions. First, the kinetochore localization of Mps1 kinase (and Bub1, Plk1 kinase and CDK1-Cyclin B) positions the kinase close to their substrates (i.e., Knl1). Second, the kinetochore localization of Bub1 serves as a scaffold to recruit its downstream factors such as BubR1, Mad1/Mad2 and RZZ. Last, the kinetochore localization of Mps1 and Bub1 may facilitate their own activation due to the higher local concentration at kinetochore.

The hierarchical recruitment pathway of SAC is becoming elucidated gradually. In brief, Aurora B activity boosts the kinetochore recruitment and activation of Mps1. Then, Mps1 phosphorylates Knl1, and in turn, phosphorylated Knl1 recruits Bub1/Bub3. Bub1 works as a scaffold to recruit BubR1/Bub3, Mad1/Mad2, RZZ and Cdc20. Despite important progress, many outstanding questions remain. For example, an exact molecular delineation of how Aurora B activity and ARHGEF17 promote Mps1 kinetochore recruitment remains elusive. Future studies to address these questions will definitely deepen our understanding on SAC signaling. Advanced protein structural analyses, protein-protein interaction interface delineation and protein localization dynamics analyses using super-resolution imaging tool combination with optogenetic operation will pave our way in future.

## Figures and Tables

**Figure 1 cells-08-00278-f001:**
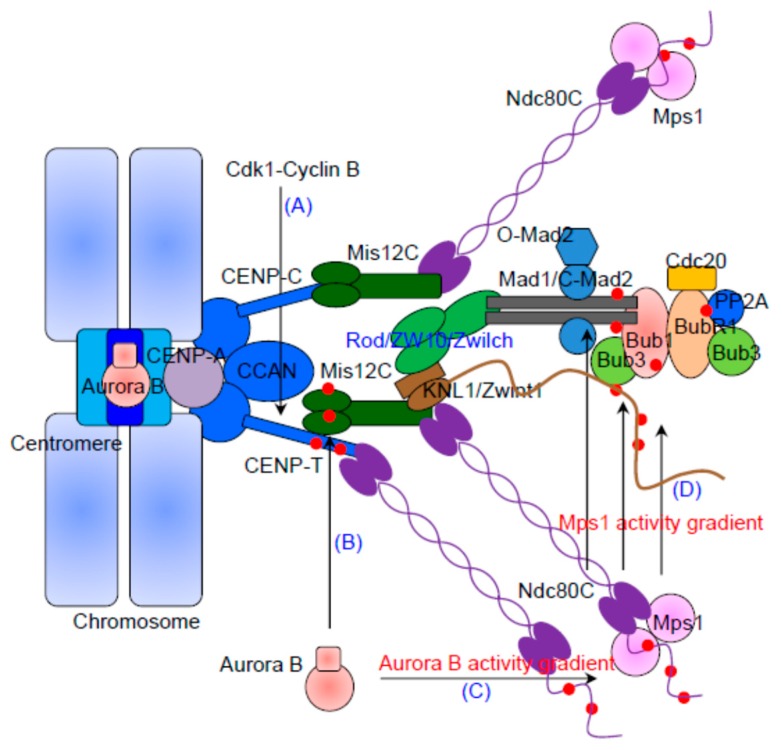
The Knl1-Mis12-Ndc80 (KMN) network recruits spindle assembly checkpoint (SAC) proteins to the kinetochores without (correct) microtubule attachment. The diagram shows the kinetochore key components assembly in early mitosis. Upon mitotic entry, the Cdk1-Cyclin B phosphorylation of multiple substrates promotes outer kinetochore assembly. Specifically, the Cdk1 phosphorylation towards CENP-T significantly enhances the binding of Ndc80C to CENP-T (**A**). Aurora B phosphorylation towards multiple substrates also plays a crucial role in kinetochore assembly and SAC factors recruitment. It phosphorylates Mis12C and enhances the loading of Mis12C to CENP-T (**B**). Most importantly, the Aurora B phosphorylation of Ndc80/Hec1 N-tail plays dual roles: destabilizing microtubule attachment and enhancing Mps1 recruitment (**C**). Once activated, Mps1 can phosphorylate Knl1. Then, phosphorylated Knl1 targets the Bub1-Bub3 complex, and then, Bub1 acts as a scaffold to recruit BubR1-Bub3 and Mad1-Mad2. Mps1 phosphorylates Bub1 and Mad1 subsequently to boost Mad1-Mad2 kinetochore recruitment (**D**). We note that there are multiple phosphorylation-mediated protein docking/assembly events on kinetochores, and we only labeled the key events highly related with this review.

**Figure 2 cells-08-00278-f002:**
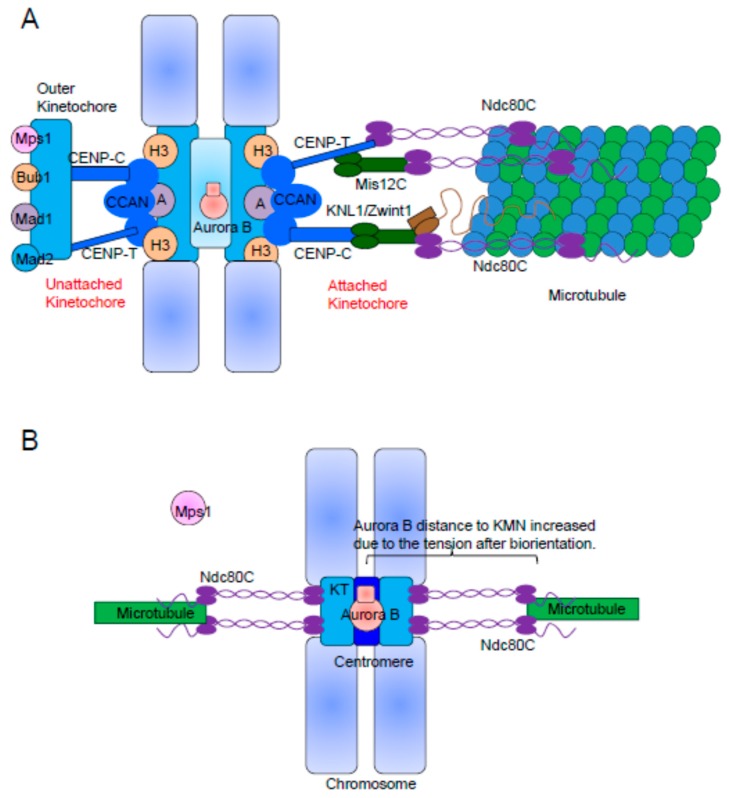
The molecular models to couple microtubule attachment with SAC signaling: (**A**) A model of the competitive binding to Ndc80C between microtubule and Mps1 kinase. Due to the competitive binding with Ndc80C, the microtubule attachment leads to the dissociation of Mps1 from the kinetochore. As a consequence, PP1 and PP2A dephosphorylate Mps1 substrates Knl1, Bub1 and Mad1. SAC signaling is quenched quickly. (**B**) A model of the spatial separation of Aurora B with the outer kinetochore substrates upon tension establishment. Upon the establishment of bipolar spindle attachment, Aurora B is unable to phosphorylate its outer kinetochore substrates (most importantly, Hec1) due to tension-caused spatial separation. Without the persistent phosphorylation by Aurora B, dephosphorylated Hec1 results in the dissociation of Mps1 and, finally, the silence of SAC signaling.

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
