# Peer review of "Recent Progress on the Localization of the Spindle Assembly Checkpoint Machinery to Kinetochores"

_cells, 2019, doi:10.3390/cells8030278_

Round 1

Reviewer 1 Report

The spindle assembly checkpoint (SAC) is a surveillance system that ensures fidelity in chromosome segregation in eukaryotes. Although SAC genes were identified more than 25 years ago, it remained unclear until recently how SAC proteins function at molecular levels. Recent years witnessed a number of important findings from various labs including the authors’ labs. This well-written manuscript highlights recent findings in a timely manner and should be useful for people in the field. I support its rapid publication in Cells.

Suggestions

-       Introduction is quite long, is somewhat redundant with other sections, and includes some of latest findings which should be moved to later sections (e.g. references 31–33). The authors may streamline the Introduction to make the manuscript easier to read. It would also be better if they introduce kinetochores (mentioned in line 112 in this manuscript) in the Introduction before mentioning, for example, Ndc80 N-terminal tail (line 43).

-       The authors may add a dedicated section that explains why kinetochore localization of SAC components are critical for their function. It is scattered in multiple places in this manuscript (e.g. line 88, line 108).

-       Line 63: “has acquired” should be corrected to “has” because the ancient Madbub molecule is thought to have had an active kinase domain, which in vertebrate became a pseudokinase.

-       Line 145: The authors may consider citing the recent Hara et al, paper (https://doi.org/10.1038/s41556-018-0230-0).

-       Line 133: Hec16A should be Hec1-6A

-       Line 226: Hec1-9A, Hec1-9D

-       Line 23: “whether”

-       Figure 1: “CCAN” near the left centromere should be removed.

-       Figure 2A: This figure is not very clear at a glance. The authors may include another kinetochore that is not bound to MTs to better explain the model.

Author Response

Journal: Cells

Manuscript Status: Pending minor revisions

Manuscript ID: cells-456016

Type: Review

Title: Recent progress into kinetochore localization of spindle assembly checkpoint signaling machinery

Referee 1

The spindle assembly checkpoint (SAC) is a surveillance system that ensures fidelity in chromosome segregation in eukaryotes. Although SAC genes were identified more than 25 years ago, it remained unclear until recently how SAC proteins function at molecular levels. Recent years witnessed a number of important findings from various labs including the authors’ labs. This well-written manuscript highlights recent findings in a timely manner and should be useful for people in the field. I support its rapid publication in Cells.

We thank the reviewer for his/her enthusiasm, support and constructive suggestions. We have taken his/her inputs and carried out a careful revision of our manuscript.

Suggestions

-   Introduction is quite long, is somewhat redundant with other sections, and includes some of latest findings which should be moved to later sections (e.g. references 31–33). The authors may streamline the Introduction to make the manuscript easier to read. It would also be better if they introduce kinetochores (mentioned in line 112 in this manuscript) in the Introduction before mentioning, for example, Ndc80 N-terminal tail (line 43).

We thank the great suggestion of reviewer. We now introduced the kinetochore ahead of the SAC in first paragraph. We removed the sentences related with references 31-33 to part 5 in our revised manuscript since kinetochore expansion is relevant with RZZ complex.

-   The authors may add a dedicated section that explains why kinetochore localization of SAC components are critical for their function. It is scattered in multiple places in this manuscript (e.g. line 88, line 108).

We thank the great suggestion. We combined the content of the importance of kinetochore recruitment of SAC components in the third paragraph in “Introduction” part in revised manuscript.

-   Line 63: “has acquired” should be corrected to “has” because the ancient Mad-bub molecule is thought to have had an active kinase domain, which in vertebrate became a pseudokinase.

We thank the great suggestion of reviewer. We deleted “acquired”.

-  Line 145: The authors may consider citing the recent Hara et al, paper (https://doi.org/10.1038/s41556-018-0230-0).

We thank the great suggestion of reviewer. We described the new findings by Hara et al., in our revised version and emphasized the phosphorylation-regulated kinetochore plasticity.

-   Line 133: Hec16A should be Hec1-6A

We apologize for the confusion. The mistake had been changed.

-   Line 226: Hec1-9A, Hec1-9D

We apologize for the confusion. The mistake had been changed.

-   Line 233: “whether”

We apologize for the typo. We changed “weather” as “whether”.

-   Figure 1: “CCAN” near the left centromere should be removed.

Thanks for the suggestion. We had removed the “CCAN” near the left centromere.

-  Figure 2A: This figure is not very clear at a glance. The authors may include another kinetochore that is not bound to MTs to better explain the model.

Thanks for the great suggestion. We now include another kinetochore not bound to microtubule.

Reviewer 2 Report

p.p1 {margin: 0.0px 0.0px 0.0px 0.0px; line-height: 15.0px; font: 13.3px Arial; color: #000000; -webkit-text-stroke: #000000; background-color: #ffffff} p.p2 {margin: 0.0px 0.0px 0.0px 0.0px; line-height: 15.0px; font: 13.3px Arial; color: #000000; -webkit-text-stroke: #000000; background-color: #ffffff; min-height: 15.0px} span.s1 {font-kerning: none}

This review by Dou et al. is a concise update on spindle assembly checkpoint signaling that will be a useful resource for researchers in the field. The manuscript requires copyediting, since there are spelling and grammatical errors within the paper.  I suggest publication after addressing the following points (each point is listed in the same order as the manuscript) to improve clarity, accuracy, and readability by a wider audience:

1) The title should read “Recent progress on the localization of the spindle assembly checkpoint machinery to kinetochores” 

2) In the abstract, line 22, the word “evolved” is used.  It’s not clear why this is used here and it should be removed.

3) Lines 76-78….these sentences should be re-written so that the reader is only referred to other reviews once.

4) Line 92, the mention of forced Mad1 localization here seems like a non-sequitur.  The reason for mentioning this here should be made clear. How does this support the thesis of the paragraph?  

5) Lines 101-111.  Again the thesis of this paragraph seems somewhat confusing.  The thesis is that SAC component localization to kinetochores is important for the checkpoint.  However, there is quite a bit of information here about kinetochore-independent SAC signaling. The way it is written makes it sound like the interphase localization of SAC proteins is at odds with the kinetochore localization.  I feel like this section should be re-written so that the cell cycle dependence of SAC localization in interphase and mitosis is more clear. 

6) Line 123: The following reference should be included when discussing the synergistic microtubule binding of the KMN network, since it shows this explicitly.

Welburn, J. P. I., Vleugel, M., Liu, D., Yates, J. R., Lampson, M. A., Fukagawa, T., & Cheeseman, I. M. (2010). Aurora B phosphorylates spatially distinct targets to differentially regulate the kinetochore-microtubule interface. Molecular Cell, 38(3), 383–392. http://doi.org/10.1016/j.molcel.2010.02.034

7) Line 133:  Should read Hec1-6A mutant, not Hec16A mutant.

8) Line 158:  CENP-T also binds to the Mis12 Complex in a CDK-dependent manner.  This should be mentioned in the text and incorporated into the figure. The following references should be added:

Huis In 't Veld, P. J., Jeganathan, S., Petrovic, A., Singh, P., John, J., Krenn, V., et al. (2016). Molecular basis of outer kinetochore assembly on CENP-T. eLife, 5, 576. http://doi.org/10.7554/eLife.21007

Rago, F., Gascoigne, K. E., & Cheeseman, I. M. (2015). Distinct Organization and Regulation of the Outer Kinetochore KMN Network Downstream of CENP-C and CENP-T. Current Biology : CB, 25(5), 671–677. http://doi.org/10.1016/j.cub.2015.01.059

9) Section 3 (begins with line 176):  Is the role of Mps1 in error correction mentioned here? I couldn’t find it.  If not, recent work into this topic should be referenced (see one example below) in one or two sentences so that the reader is clear that it has other functions.

Mps1 Regulates Kinetochore-Microtubule Attachment Stability via the Ska Complex to Ensure Error-Free Chromosome Segregation. (2017). Mps1 Regulates Kinetochore-Microtubule Attachment Stability via the Ska Complex to Ensure Error-Free Chromosome Segregation., 41(2), 143–156.e6. http://doi.org/10.1016/j.devcel.2017.03.025

10) Line 211:  The localization of Mps1 to kinetochores in the presence of a related inhibitor, STLC, was also observed in Xenopus egg extracts and should be referenced.  The idea that syntelic attachments might not all be end-on should be referenced as well.  This would also be a good place to suggest that the role of Mps1 in chromosome alignment and error correction may require different factors.

Haase, J., Bonner, M. K., Halas, H., & Kelly, A. E. (2017). Distinct Roles of the Chromosomal Passenger Complex in the Detection of and Response to Errors in Kinetochore-Microtubule Attachment. Developmental Cell, 42(6), 640–654.e5. http://doi.org/10.1016/j.devcel.2017.08.022

Shrestha, R. L., & Draviam, V. M. (2013). Lateral to end-on conversion of chromosome-microtubule attachment requires kinesins CENP-E and MCAK. Current Biology : CB, 23(16), 1514–1526. http://doi.org/10.1016/j.cub.2013.06.040

11) Line 226:  Should read Hec1-9A and Hec1-9D

12) Line 225:  The work by Nijenhuis et al. demonstrates that Aurora B is regulating Mps1 directly to promote its kinetochore localization. This should be made explicit so that the reader can understand that there were other mechanisms proposed and not just a simple disagreement in results.

13) Line 227: Haase et al. (see reference listed above in point 10) present evidence that centromeric Aurora is not required for checkpoint recruitment of BubR1 and Mad2 to kinetochores when attachments are lost.  This is evidence against the spatial-separation of Aurora B being important for the SAC and should be mentioned.

14) Line 274:  More details should be given about the KI motifs in KNL1.  I don’t think a person outside the field would understand what they are.

15) Paragraph starting on Line 305:  This paragraph needs to be re-written for clarity.  I feel like moving the sentence from line 330-332 to the beginning of the paragraph would be helpful for the reader.  I was especially confused at Line 316, “it is therefore believed…” because it was unclear how many pathways were being discussed here.

Author Response

Journal: Cells

Manuscript Status: Pending minor revisions

Manuscript ID: cells-456016

Type: Review

Title: Recent progress into kinetochore localization of spindle assembly checkpoint signaling machinery

Referee 2

This review by Dou et al. is a concise update on spindle assembly checkpoint signaling that will be a useful resource for researchers in the field. The manuscript requires copyediting, since there are spelling and grammatical errors within the paper.  I suggest publication after addressing the following points (each point is listed in the same order as the manuscript) to improve clarity, accuracy, and readability by a wider audience:

Thank the reviewer for his/her enthusiasm, support and constructive suggestions. We have taken his/her inputs and carried out a careful revision of our manuscript. We believe the revised manuscript improved greatly.

1) The title should read “Recent progress on the localization of the spindle assembly checkpoint machinery to kinetochores”

 We thank the reviewer for his/her great suggestion and we adopted it.

2) In the abstract, line 22, the word “evolved” is used.  It’s not clear why this is used here and it should be removed.

Thanks for great suggestion. We removed “evolved” in current manuscript.

3) Lines 76-78….these sentences should be re-written so that the reader is only referred to other reviews once.

We re-wrote the corresponding sentences.

4) Line 92, the mention of forced Mad1 localization here seems like a non-sequitur.  The reason for mentioning this here should be made clear. How does this support the thesis of the paragraph?

Thanks for great suggestion! We deleted the sentences describing forced Mad1 kinetochore localization in current manuscript.

5) Lines 101-111.  Again the thesis of this paragraph seems somewhat confusing.  The thesis is that SAC component localization to kinetochores is important for the checkpoint. However, there is quite a bit of information here about kinetochore-independent SAC signaling. The way it is written makes it sound like the interphase localization of SAC proteins is at odds with the kinetochore localization. I feel like this section should be re-written so that the cell cycle dependence of SAC localization in interphase and mitosis is more clear.

Thanks for great suggestion! We re-wrote the corresponding sentences. It is widely appreciated that the kinetochore localization of SAC factors is of great importance. Nevertheless, evidence exist that APC/C inhibitor was generated in interphase, independent of the kinetochore localization of SAC factors. Therefore, to give the readers across-the-board understanding, we described the presence of kinetochore-independent SAC signaling.

6) Line 123: The following reference should be included when discussing the synergistic microtubule binding of the KMN network, since it shows this explicitly.

Welburn, J. P. I., Vleugel, M., Liu, D., Yates, J. R., Lampson, M. A., Fukagawa, T., & Cheeseman, I. M. (2010). Aurora B phosphorylates spatially distinct targets to differentially regulate the kinetochore-microtubule interface. Molecular Cell, 38(3), 383–392. http://doi.org/10.1016/j.molcel.2010.02.034

We apologize that we neglected the important literature the reviewer mentioned. This reference was added in our revised manuscript.

7) Line 133:  Should read Hec1-6A mutant, not Hec16A mutant.

We apologize for the confusion. The mistake had been changed.

8) Line 158:  CENP-T also binds to the Mis12 Complex in a CDK-dependent manner.  This should be mentioned in the text and incorporated into the figure. The following references should be added:

Huis In 't Veld, P. J., Jeganathan, S., Petrovic, A., Singh, P., John, J., Krenn, V., et al. (2016). Molecular basis of outer kinetochore assembly on CENP-T. eLife, 5, 576. http://doi.org/10.7554/eLife.21007

Rago, F., Gascoigne, K. E., & Cheeseman, I. M. (2015). Distinct Organization and Regulation of the Outer Kinetochore KMN Network Downstream of CENP-C and CENP-T. Current Biology : CB, 25(5), 671–677. http://doi.org/10.1016/j.cub.2015.01.059

We apologize that we neglected the important literatures the reviewer mentioned. These references were added in our revised manuscript.

9) Section 3 (begins with line 176):  Is the role of Mps1 in error correction mentioned here? I couldn’t find it.  If not, recent work into this topic should be referenced (see one example below) in one or two sentences so that the reader is clear that it has other functions.

Mps1 Regulates Kinetochore-Microtubule Attachment Stability via the Ska Complex to Ensure Error-Free Chromosome Segregation. (2017). Developmental Cell 41(2), 143–156.e6. http://doi.org/10.1016/j.devcel.2017.03.025

Thanks for great suggestion! We mentioned the role of Mps1 in error-correction and cited the reference as suggested.

10) Line 211:  The localization of Mps1 to kinetochores in the presence of a related inhibitor, STLC, was also observed in Xenopus egg extracts and should be referenced.  The idea that syntelic attachments might not all be end-on should be referenced as well. This would also be a good place to suggest that the role of Mps1 in chromosome alignment and error correction may require different factors.

Haase, J., Bonner, M. K., Halas, H., & Kelly, A. E. (2017). Distinct Roles of the Chromosomal Passenger Complex in the Detection of and Response to Errors in Kinetochore-Microtubule Attachment. Developmental Cell, 42(6), 640–654.e5. http://doi.org/10.1016/j.devcel.2017.08.022

Shrestha, R. L., & Draviam, V. M. (2013). Lateral to end-on conversion of chromosome-microtubule attachment requires kinesins CENP-E and MCAK. Current Biology : CB, 23(16), 1514–1526. http://doi.org/10.1016/j.cub.2013.06.040

Thanks for the references suggested! This is a great evidence to support our point of view. We argue the idea that syntelic attachments might not all be end-on. We also described the role of Mps1 in chromosomal alignment and cited the reference as suggested.

11) Line 226:  Should read Hec1-9A and Hec1-9D

We apologize for the confusion. The mistake had been changed.

12) Line 225: The work by Nijenhuis et al. demonstrates that Aurora B is regulating Mps1 directly to promote its kinetochore localization. This should be made explicit so that the reader can understand that there were other mechanisms proposed and not just a simple disagreement in results.

Thanks for great suggestion! We re-wrote the related content and emphasize the presence of alternative mechanisms.

13) Line 227: Haase et al. (see reference listed above in point 10) present evidence that centromeric Aurora is not required for checkpoint recruitment of BubR1 and Mad2 to kinetochores when attachments are lost.  This is evidence against the spatial-separation of Aurora B being important for the SAC and should be mentioned.

Thanks for great suggestion! We described this interesting finding and mentioned it is against the spatial separation of Aurora B model.

14) Line 274: More details should be given about the KI motifs in KNL1. I don’t think a person outside the field would understand what they are.

Thanks for great suggestion! We added the following description for KI motif: short 12-residue motifs, termed Lys-Ile.

15) Paragraph starting on Line 305:  This paragraph needs to be re-written for clarity.  I feel like moving the sentence from line 330-332 to the beginning of the paragraph would be helpful for the reader.  I was especially confused at Line 316, “it is therefore believed…” because it was unclear how many pathways were being discussed here.

Thanks for great suggestion! We re-wrote this paragraph in revised manuscript. “it is therefore believed…” was deleted and re-written. We believe that current version improves greatly.